biotechnology/bioengineering/health and disease and epidemiology

endoscope, inflammatory bowel disease, electrical impedance, biomarkers, colitis

**Author for correspondence:**
Sophie C. Payne
e-mail: spayne@bionicsinstitute.org

# Transmural impedance detects graded changes of inflammation in experimental colitis

Sophie C. Payne[1,2], Jack Alexandrovics[1], Ross Thomas[1], Robert K. Shepherd[1,2], John B. Furness[3,5] and James B. Fallon[1,2,4]

[1]Bionics Institute, Fitzroy, Victoria 3065, Australia
[2]Medical Bionics Department, [3]Department of Anatomy and Neuroscience, and [4]Department of Otolaryngology, the University of Melbourne, Parkville, Victoria 3010, Australia
[5]Florey Institute of Neuroscience and Mental Health, Parkville, Victoria, Australia

 SCP, 0000-0002-3428-2275; RKS, 0000-0002-4239-3362;
JBFu, 0000-0002-0219-3438; JBFa, 0000-0003-2686-3886

Ulcerative colitis is a chronic disease in which the mucosa of the colon or rectum becomes inflamed. An objective biomarker of inflammation will provide quantitative measures to support qualitative assessment during an endoscopic examination. Previous studies show that transmural electrical impedance is a quantifiable biomarker of inflammation. Here, we hypothesize that impedance detects spatially restricted areas of inflammation, thereby allowing the distinction between regions that differ in their severity of inflammation. A platinum ball electrode was placed into minimally inflamed (i.e. normal) or 2,4,6-trinitrobenzene sulphonic acid (TNBS)-inflamed colonic regions of rats and impedance measurements obtained by passing current between the intraluminal and subcutaneous return electrode. Histology of the colon was correlated with impedance measurements. The impedance of minimally inflamed (normal) tissue was 1.5–1.9 kΩ. Following TNBS injection, impedance significantly decreased within the inflammatory penumbra ($p < 0.05$), and decreased more in the inflammatory epicentre ($p = 0.02$). Histological damage correlated with impedance values ($p < 0.05$). Thus, impedance values of 1.5–1.9, 1.3–1.4 and 0.9–1.1 kΩ corresponded to minimally inflamed, mildly inflamed and moderately inflamed tissue, respectively. In conclusion, transmural impedance is an objective, spatially localized biomarker of mucosal integrity, and distinguishes between severities of intestinal inflammation.

# 1. Introduction

Ulcerative colitis (UC) is an inflammatory bowel disease (IBD) that affects parts of, or the entire, colon and rectum. This progressive, but episodic, disease often starts in young adulthood and affects patients throughout their lives [1]. The debilitating symptoms of the disease include weight loss, diarrhoea, haemorrhage, perforation and dysplastic lesions [2] that may lead to a colectomy in some patients [3].

UC is diagnosed in patients by assessing clinical symptoms and biological markers of inflammation, including endoscopic appearance, histological damage in biopsies of affected tissue and cytokine measurements [2,4]. Endoscopic grading systems, such as the Baron Score or Modified Baron Score, are currently used for clinical assessment of UC [5]. However, these scoring systems have a subjective component, and inter-observer variability in grading disease activity has previously been reported [6–8]. The addition of an observer-independent method of grading UC would allow for improved diagnosis and monitoring of the disease.

A reliable way to measure inflammation of the mucosa is to assess alterations to intestinal permeability [9]. When the intestine is inflamed, the trans-mucosal permeability is increased due to the loss of epithelial cells and/or the disruption of tight junctions between epithelial cells [9,10]. Recently, we demonstrated that inflammation-induced disruption to mucosal integrity can be monitored in real time, by measuring changes in transmural electrical impedance between electrodes placed within the small intestine and an external return electrode [10]. The changes to transmural impedance correlated with histologically assessed damage to the mucosa and infiltration of leucocytes, which are established measurements of inflammation. However, our previous study was conducted in the small intestine and it was necessary to surgically implant the electrodes into the lumen, thereby penetrating the wall of the intestine. Here, we investigated the colon and introduced the electrodes via the rectum, an approach that has greater potential for clinical application, for example during a colonoscopy. In fact, in a recent prospective clinical study [11], the mucosal integrity of the rectum of UC patients was assessed by measuring impedance (that was reported as admittance, the inverse of impedance) with a rectally introduced probe. This measurement was able to predict the prognosis for the patient, with higher impedance being associated with long-term sustained remission [11]. Thus, measuring the electrical impedance of the intestinal mucosa during an endoscopic examination could be a useful, supplementary diagnostic tool in UC [10,11].

In this previous clinical study [11], there was no assessment as to whether transmural impedance was related to the severity of inflammatory damage. Here, we aim to use an established experimental model of 2,4,6-trinitrobenzene sulphonic acid (TNBS)-induced colitis in rats [12] to assess whether changes in transmural impedance were spatially restricted to the site of inflammation and able to predict severities of intestinal inflammation. A platinum ball electrode was inserted into the colon and impedance measurements were obtained at the inflammatory epicentre and adjacent colon, and compared with measurements taken from the minimally inflamed (normal) colon by passing current between the intraluminal electrode and a return electrode located subcutaneously. Histological damage and infiltration of inflammatory cells were quantified and correlated with corresponding impedance values from the same anatomical location.

# 2. Material and methods

## 2.1. Animals and experimental colitis

All experiments used male Sprague-Dawley rats ($n = 8$ rats; 350–370 g, eight weeks old, Animal Resource Centre, Western Australia) and were approved by the Animal Research and Ethics Committee of the Bionics Institute. All experiments complied with the Australian Code for the Care and Use of Animals for Scientific Purposes (National Health and Medical Research Council of Australia). Animals were allowed ad libitum access to standard chow, water and fresh food, and were kept on a 12 h light/dark cycle. Normal rats ($n = 4$) were anaesthetized (2% isoflurane, flow rate of 1–1.5 l min$^{-1}$) and a catheter inserted via the rectum to inject 0.25 ml of sterile saline precisely at 8 cm into the colon. An impedance measurement (detailed below) was taken, the animal euthanized and tissues taken for histology. In experimental colitis rats ($n = 4$), baseline impedance measurements were taken at 5 min prior to the TNBS injection ($T = -5$ min). At $T = 0$ min, colitis was induced by injecting 0.25 ml of 1% TNBS (in 50% ethanol, Sigma) exactly 8 cm into the colon [12]. Rats were held in a head-down position for 5 min following the intra-colonic injection. At 90 min following the TNBS injection, an

impedance measurement was taken ($T = 90$ min), and animals were terminated (300 mg kg$^{-1}$ Lethabarb, intra-cardiac injection) and tissue taken for histology.

## 2.2. Impedance measurements

Impedance measurements were taken by inserting a platinum ball electrode (approximate surface area: 3.2 mm$^2$), supported by a silicone carrier, through the rectum. A subcutaneous needle acted as a return (figure 1a). Impedance measurements were taken from normal rats at 5.5 and 8 cm into the colon, from the anal margin. In the cohort of rats selected to receive the TNBS injection, a baseline internal control impedance measurement was taken (5.5 and 8 cm into the colon; figure 1bi,bii) at 5 min ($T = -5$ min) prior to the TNBS injection. At 90 min ($T = 90$ min) following the TNBS injection, impedance measurements were taken within the inflammatory penumbra (adjacent region, 5.5 cm into the colon) and inflammatory epicentre (8 cm into the colon) (figure 1a,bi,bii).

Impedance was monitored by recording the voltage transient produced by passing biphasic current pulses between the intra-colonic electrode and a subcutaneous return electrode [10]. A 25 µs per phase biphasic current pulse (7 µs interphase gap) at 931 µA was used to produce a peak voltage transient of approximately 500 mV when the platinum ball electrode was placed in saline. The peak voltage at the end of the first phase of the current pulse was recorded and used to calculate impedance using Ohm's Law ($Z =$ voltage/current) [13]. The average of three impedance measurements was used and analysed. Following the end of the experiment, the platinum ball electrode was cleaned and retested in saline.

## 2.3. Histology and staining

A sample of colonic tissue was taken 5.5 and 8 cm into the colon from normal animals. In experimental colitis animals, at 90 min following the TNBS injection rats were euthanized and colonic tissue taken at the inflammatory penumbra site (5.5 cm into the colon) and at the inflammatory epicentre (8 cm into the colon). Similar to previous studies, colonic tissue was dissected out, placed in Zamboni's fixative (2% formaldehyde plus 0.2% picric acid in 0.1 M sodium phosphate buffer, pH 7.4) overnight, embedded in paraffin and sectioned (5 µm). Slides were stained with haematoxylin and eosin (H&E) or immunohistochemistry performed using the cytotoxic T-cell marker anti-CD3 (1 : 200 in 10% normal horse serum, Dako Cytomation) and mounted with DPX [10,14].

## 2.4. Histopathology scoring

A single observer (S.C.P.), blinded to experimental conditions, used H&E stained sections to evaluate the degree of inflammation. Tissue was scored according to parameters outlined in table 1, which were adapted from previous studies [15,16]. Scores could range from 0 (no inflammation) to a maximum of 15 (severe inflammation).

## 2.5. Quantification of leucocytes

Cytotoxic T-lymphocytes [17] were identified using antibodies to CD3 as this immunoglobulin is present at all stages of T cell development. H&E sections were used to identify eosinophilic granulocytes morphologically by their distinctive nucleus and cytoplasmic staining [14]. The distinctive nuclei of banded, mature or hyper-segmented neutrophils were also identified morphologically. Positive cells were counted with ×40 objective, across 10 fields of view for each sample using a Zeiss Axioplan II microscope. Positive cells were counted within the mucosa, submucosa and smooth muscle layers and the number of cells per millimetre section length was calculated and analysed.

## 2.6. Statistical analysis

Raw colonic wall impedance values were reported and a two-way repeated measures ANOVA (Location × Time) and Sidak *post hoc* test used to test for differences and interactions between impedance measurements. Infiltration of leucocytes into the transmural layers of the minimally inflamed and inflamed colon was assessed using a one-way ANOVA and a Tukey's *post hoc* test. Impedance measurements were correlated with histological score or leucocyte infiltration using a Pearson's correlation coefficient ($r$) and two-tailed $p$-values generated. A $p$-value of $p < 0.05$ was accepted as statistically significant and data expressed as mean ± s.e.m. GraphPad Prism 4 was used for all analyses (GraphPad Software, USA).

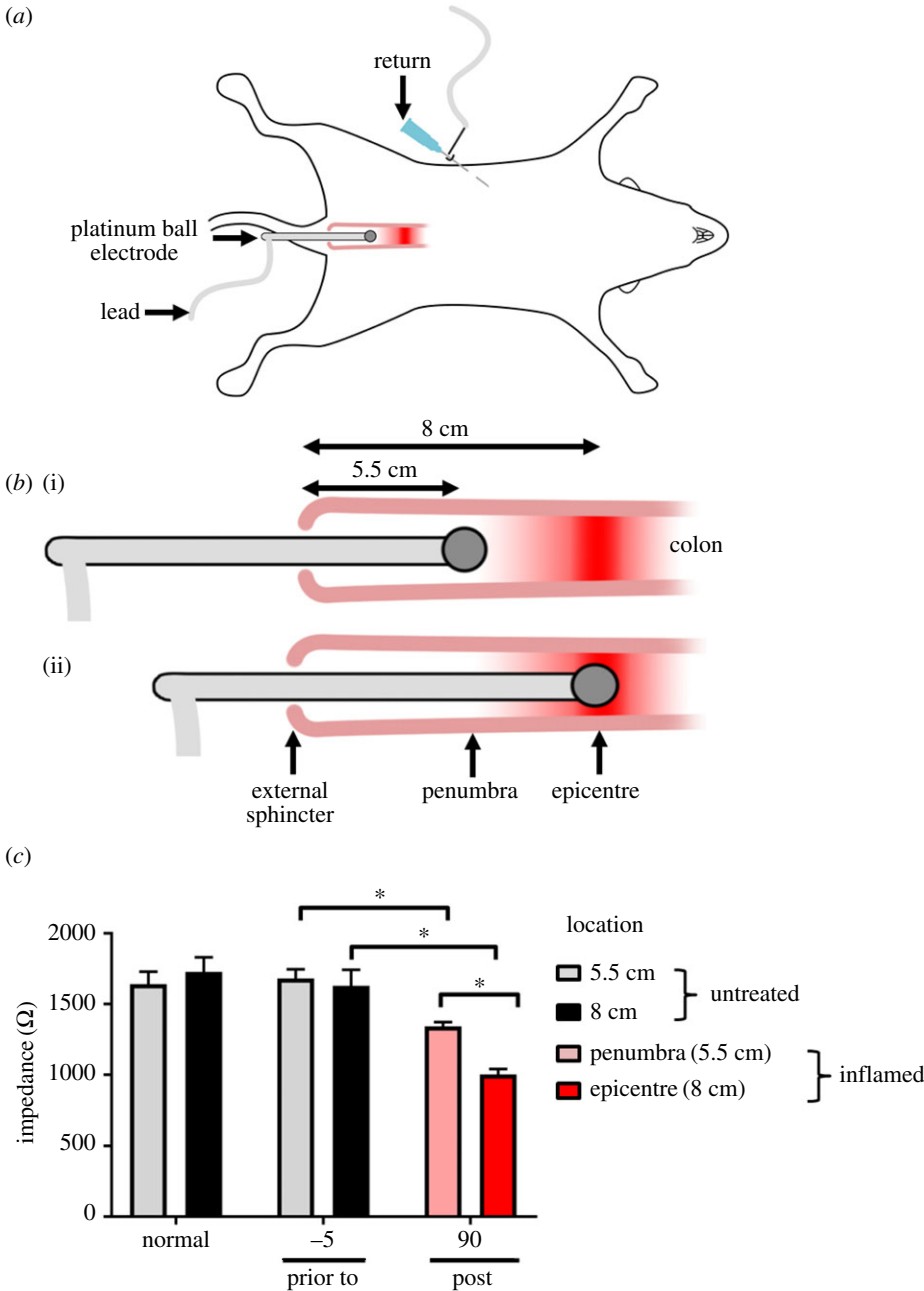

**Figure 1.** Experimental method and quantification of impedance changes following TNBS-induced colitis. (*a,b*) Schematic diagram showing experimental design (*a*). (*bi,bii*) The inflammatory agent (TNBS, 2,4,6-trinitrobenzene sulfonic acid) was injected into the colon 8 cm from the external sphincter. Impedance measurements were taken using a platinum ball electrode within the inflammatory penumbra ((*bi*), 5.5 cm from external sphincter) and at the inflammatory epicentre ((*bii*), 8 cm from the external sphincter). Prior to TNBS ($T = -5$ min) measurements were taken in the absence of TNBS at the same anatomical locations (5.5 and 8 cm). (*c*) At 90 min following the TNBS injection, impedance measurements taken within the inflammatory penumbra and epicentre decreased ($T = -5$ min; $p < 0.05$). Furthermore, at 90 min, impedance measurements taken from the inflammatory epicentre were significantly less than those taken within the penumbra ($p = 0.02$). Data show mean impedance $\pm$ s.e.m. and differences were considered significant for $p < 0.05$.

## 3. Results

### 3.1. Graded change in impedance following TNBS injection

Impedance measured from normal animals ($n = 4$) was $1.64 \pm 0.9$ kΩ (range: 1.50–1.84 kΩ; s.d.: 0.18 kΩ) and $1.73 \pm 0.1$ kΩ (range: 1.50–1.9; s.d.: 0.21 kΩ)—at 5.5 and 8 cm into the colon, respectively. There

**Table 1.** Histological scoring system of inflammatory damage within colonic tissue.

| parameter | criteria | scoring criteria | score |
|---|---|---|---|
| inflammatory changes | *severity*<br>change in leukocyte density of lamina propria area | no change | 0 |
| | | mild: 10–25% | 1 |
| | | moderate: 26–50% | 2 |
| | | marked: >51% | 3 |
| | *extent*<br>expansion of leukocyte infiltration | none | 0 |
| | | mucosal | 1 |
| | | mucosal and submucosal | 2 |
| | | transmural | 3 |
| epithelial damage | *goblet cell loss*<br>reduction of goblet cell numbers relative to baseline goblet cell numbers per crypt | none | 0 |
| | | mild: 10–25% | 1 |
| | | moderate: 26–50% | 2 |
| | | marked: >50% | 3 |
| | *erosion*<br>loss of surface epithelium across tissue | none | 0 |
| | | mild: 10–25% | 1 |
| | | moderate: 26–50% | 2 |
| | | marked: >50% | 3 |
| mucosal architecture | *extent of damage to crypts*<br>irregular crypts, crypt loss | no damage | 0 |
| | | damage affects less than 1/3 of crypts | 1 |
| | | damage affects between 1/3 and 2/3 of crypts | 2 |
| | | damage affects to more than 2/3 of crypts | 3 |

was no difference between these two anatomical locations (paired student $t$-test: $p = 0.34$). Prior to TNBS ($n = 4$), baseline impedances ($n = 4$) taken 5 min prior ($T = -5$ min) to the TNBS injection were not different to that of normal values (5.5 cm location: $1.68 \pm 0.6$ kΩ; 8 cm location: $1.63 \pm 0.1$ kΩ; unpaired $t$-test $p > 0.05$).

In experimental colitis rats, changes in impedance were tested between baseline impedance at $T = -5$ and at $T = 90$ min using a two-way RM ANOVA ($n = 4$; interaction: $p = 0.04$; time: $p < 0.0001$; location: $p = 0.13$). At 90 min following the TNBS injection, impedance measurements taken at the inflammatory penumbra (figure 1$b$i, 5.5 cm location) significantly decreased (Sidak's *post hoc* test: $p = 0.02$) to $1.34 \pm 0.3$ kΩ (range: 1.30–1.41 kΩ; s.d.: 0.06 kΩ) and impedance measurements taken at the inflammatory epicentre decreased to $1.0 \pm 0.4$ kΩ (range: 0.94–1.11 kΩ; s.d.: 0.08 kΩ; $p = 0.0001$, compared with prior TNBS measures ($T = -5$ min; figure 1$b$i,$c$). Furthermore, at 90 min impedance measurements at the inflammatory epicentre were significantly less than those taken at the inflammatory penumbra ($p = 0.02$; figure 1$c$). At the conclusion of the experiment, electrodes were cleaned and impedance measurements taken *in vitro* in sterile saline. There were no significant differences (paired $T$-test; $p \geq 0.05$) between the impedance of the Pt ball measured *in vitro* pre- and post the experiment, suggesting there was no change to the electrode's intrinsic electrical properties and the changes to the measured impedance were due to *in vivo* changes associated with the tissue.

## 3.2. Histological inflammatory damage correlates with impedance

In normal (minimally inflamed) colonic tissue, no clusters of leucocytes were observed within the tissue. Goblet cells were observed within crypts and the epithelial layer, which was intact and undamaged, and crypts had a normal architecture (figure 2$a$,$a$i).

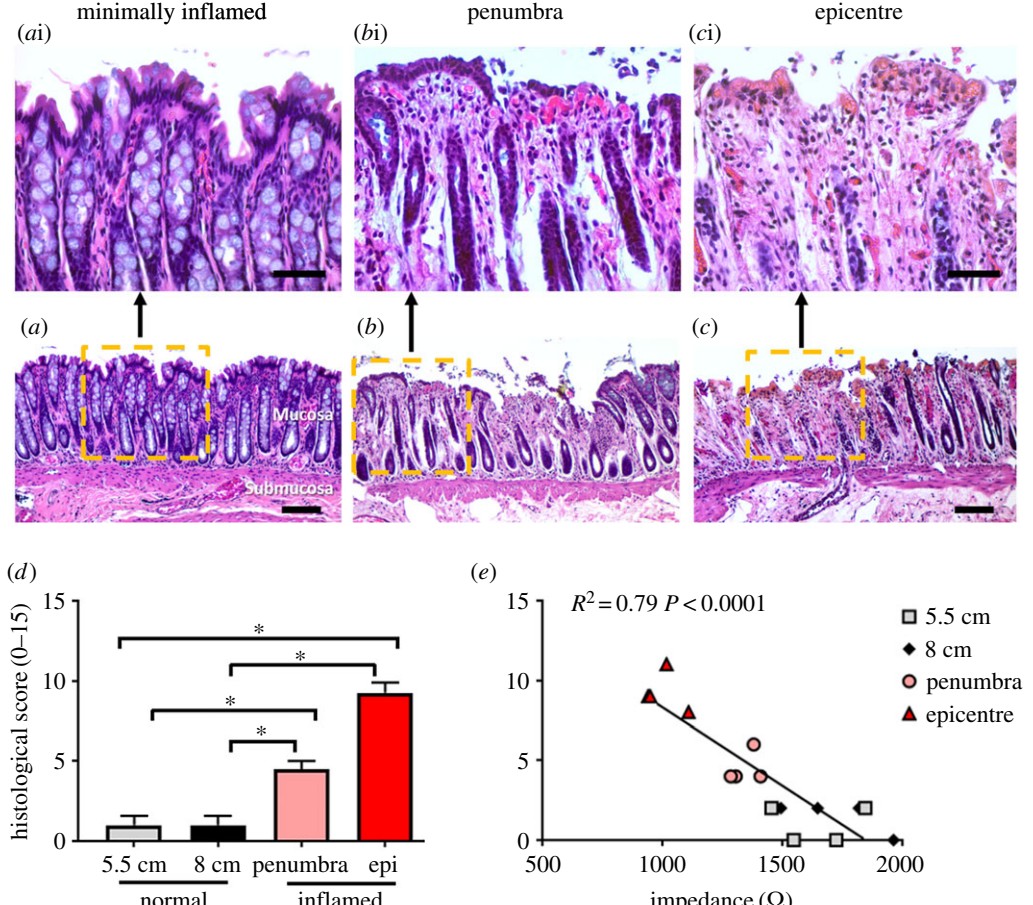

**Figure 2.** Histological damage following TNBS injection. (*a–c*) Representative images of H&E stained sections of the colon from normal tissue (*a*), tissue from the inflammatory penumbra (*b*) and epicentre (*c*). (*d*) There was significantly more histological damage in tissue from the inflammatory penumbra ($p = 0.005$) and epicentre ($p < 0.0001$), compared with normal. Tissue from the inflammatory epicentre had more histological damage than tissue from the penumbra ($p = 0.0004$). (*e*) Inflammatory score of histological damage in the mucosa correlated with corresponding impedance measurements ($p < 0.0001$; $R^2 = 0.79$). Data show the mean histological score ± s.e.m. Scale bars in (*a–c*) represent 500 µm; (*ai–ci*) represent 50 µm.

In tissue within the inflammatory penumbra, mild leucocyte infiltration was observed within the mucosal layer, but there was an absence of leucocyte infiltration within submucosal and muscle layers. The mucosal surface had some loss of epithelial cells, and mild goblet cell loss was observed within the mucosa surface. However, the distribution of goblet cells within crypts was mostly normal and there was only minor disruption or loss of crypt architecture (figure 2*b,b*i).

Tissue taken from the inflammatory epicentre had mild leucocyte infiltration within the mucosal layer, while submucosal and muscle layers had no infiltration. Extensive loss of epithelial cells was observed across the majority of the mucosal surface. Marked loss of goblet cells was observed within the surface epithelial cell layer, and some loss of goblet cells within crypts was also observed. Furthermore, a moderate number of crypts were lost (figure 2*c,c*i).

Quantitative scoring, based on parameters described in table 1, showed a significant increase (one-way ANOVA, Tukey's *post hoc*; $p < 0.0001$) in histological damage in tissue within the inflammatory penumbra (5.5 cm; $p = 0.005$) and epicentre (8 cm; $p < 0.0001$), compared with respective normal tissue locations (figure 2*d*). Furthermore, histological damage was significantly worse at the inflammatory epicentre (8 cm; $p = 0.0004$), compared with tissue within the penumbra (5.5 cm; figure 2*d*). Histological score of minimally inflamed (normal) and inflamed tissue was significantly correlated with impedance measurements taken from the same area (Pearson's correlation: $r = -0.89$; $p < 0.0001$; figure 2*d*). This suggests that impedance measurements detected different severities of inflammation between the inflammatory penumbra and epicentre. Specifically, impedance values above 1.5 kΩ were a predictor of minimally inflamed i.e. normal tissue (histology score range 0–2); impedance values between 1.3 and

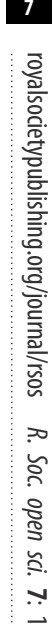

**Figure 3.** Correlation of leucocyte infiltration and impedance following TNBS injection. (*a*) Numbers of CD3+ cells per mm length of section increased in the penumbra and epicentre tissue ($p < 0.05$). CD3+ cells were higher in tissue taken from the inflammatory epicentre than at the penumbra site ($p = 0.03$). (*b*) CD3+ cell infiltration in normal and differentially inflamed tissue correlated with impedance ($R^2 = 0.57$; $p = 0.0007$). (*c*) Numbers of eosinophils increased within tissue taken from the inflammatory epicentre ($p < 0.05$). (*d*) There was a significant correlation between numbers of eosinophils within normal and differentially inflamed tissue and corresponding impedance measurements ($R^2 = 0.79$; $p < 0.0001$). (*e*) Numbers of neutrophils increased within tissue taken from the inflammatory epicentre ($p = 0.02$). (*f*) There was a significant correlation between neutrophil counts and impedance within normal and differentially inflamed tissue ($R^2 = 0.76$; $p < 0.0001$). Data show mean cell count ± s.e.m. Significant differences were accepted at $p < 0.05$.

1.4 kΩ were a predictor of mild inflammation (histology score range: 4–6); and impedance values of 0.9–1.1 kΩ were a predictor of moderately inflamed tissue (histology score range: 8–11).

## 3.3. Leucocyte cell infiltration following TNBS injection

Following the TNBS injection, numbers of CD3+ cells increased (one-way ANOVA: $p < 0.0001$) in the mucosa of tissue taken from the inflammatory penumbra ($p = 0.007$), compared with levels seen in normal tissue at the 5.5 cm location, and also increased in the inflammatory epicentre ($p = 0.0001$), compared with normal tissue at the 8 cm location. Furthermore, there were more CD3+ cells in tissue taken from the inflammatory epicentre, compared with that seen in the penumbra ($p = 0.03$; figure 3*a*). Very few CD3+ cells were observed within the submucosal layers in normal tissue (5.9 ± 1.8 CD3+ cells mm⁻¹), and there were no significant differences seen following the TNBS injection ($p \geq 0.05$).

A negligible number of CD3+ cells were seen in the muscle layers of all examined tissue. There was a significant correlation between numbers of resident mucosal CD3+ cells present and impedance values taken from the same area (Pearson's correlation: $r = -0.76$, $p = 0.0007$; figure 3$b$).

Following the TNBS injection, numbers of eosinophils in the mucosa increased (one-way ANOVA: $p = 0.0002$) in tissue taken from the inflammatory epicentre, compared with levels seen in normal tissue at the 8 cm location ($p = 0.0004$) and penumbra tissue ($p = 0.003$; figure 3$c$). There was no difference in eosinophil counts between tissue taken from the inflammatory penumbra and normal tissue at the 5.5 cm location ($p = 0.57$). Modest numbers of eosinophils observed in the submucosa significantly increased (one-way ANOVA $p = 0.02$) only in tissue taken from the inflammatory epicentre ($0.4 \pm 0.2$ cells mm$^{-1}$; $p = 0.01$), compared with normal tissue (8 cm; $0 \pm 0$ cells mm$^{-1}$). Very few eosinophils were observed within the muscle layers in normal tissue ($1 \pm 0.6$ eosinophils mm$^{-1}$) or following TNBS injection ($p = 0.43$; penumbra tissue: $0.4 \pm 0.3$ eosinophils mm$^{-1}$; epicentre tissue: $1.1 \pm 0.4$ eosinophils mm$^{-1}$). In the mucosal layer, the number of eosinophils within normal tissue, and tissue taken from the inflammatory penumbra and epicentre was significantly correlated with impedance measurements taken from the same area (Pearson's correlation: $r = -0.89$, $p < 0.0001$; figure 3$d$).

The final leucocyte population counted was the neutrophils population. TNBS injection resulted in a significant increase (one-way ANOVA: $p < 0.0001$) in neutrophils in the mucosal layer in tissue taken from the inflammatory penumbra ($p = 0.003$), compared with normal tissue at the 5.5 cm location, and with the inflammatory epicentre ($p = 0.0002$), compared with normal tissue at the 8 cm location (figure 3$e$). There was an increase ($p = 0.009$) in the number of neutrophils seen within the submucosal layer of tissue taken from the inflammatory epicentre, compared with normal tissue (8 cm) ($p = 0.01$: normal tissue: 0 neutrophils mm$^{-1}$ versus epicentre tissue: $8.8 \pm 2.8$ neutrophils mm$^{-1}$). A negligible number of neutrophils were seen in the muscle layers of all examined tissue. There was a significant correlation between mucosal neutrophil counts and impedance (Pearson's correlation: $r = -0.87$, $p < 0.0001$; figure 3$f$).

# 4. Discussion

Although the endoscopic examination is the gold standard diagnostic tool for UC, categorizing the severity of inflammatory-induced histological changes is subjective and inter-observer variability of scoring can occur [6–8]. As such, there is clinical value for a complementary, objective, method to characterize mucosal damage during the endoscopic examination. We have previously demonstrated that the change in transmural impedance provides a real-time, *in vivo* quantitative measurement of mucosal integrity [10]. In the present study, we expanded our previous observations to establish that measuring the impedance of inflamed tissue reflected localized damage, and that we were able to distinguish between the severities of inflammation in the same animal. We showed that transmural impedance was accurate in predicting the severity of inflammation as determined by histological score and leucocyte infiltration (T cells, eosinophils and neutrophils) into mucosal tissue. Thus, transmural impedance is a region specific, real-time *in vivo* indicator of mucosal integrity that distinguishes between regions with different severities of intestinal inflammation within the same animal.

We related the inflammatory score to impedance (figure 2$e$). Measuring impedance of inflamed tissue reflected localized damage that could be distinguished from the less inflamed region in the same animal. In our experiments, the measurement of the mildly inflamed region was closer to the anus (5.5 cm from the anal margin), compared with the moderately inflamed region (8 cm from the anus). However, there were no significant differences in impedance measurements taken 5.5 and 8 cm into the minimally inflamed (normal) colon. This suggests that the depth of the electrode did not affect impedance measurements, which were probably dominated by the resistive epithelial layer of the mucosa [10]. Furthermore, there is little variability between animals in impedance values generated from minimally, mildly and moderately inflamed tissue, suggesting impedance is a reliable method of examining mucosal integrity between subjects. Furthermore, changes in trans-mucosal impedance were spatially restricted to the site of inflammation and therefore able to determine both the site and the severity of intestinal inflammation. We predict that having the ability to distinguish between different levels of inflammation within the gastrointestinal tract will support the use of this diagnostic tool during an endoscopic examination.

Current clinical methods of monitoring IBD, such as macroscopic observation (endoscopy) and histological analysis of biopsies, are subjective and require significant training of the observer to perform the test accurately [9,18]. If the application of transmural impedance is implemented into an

endoscopic examination, we propose that the present technique would provide an objective, real-time biomarker that would contribute to an improved grading of disease activity. Furthermore, unlike endoscopic examination scoring systems, this technology is objective, easy to use and does not require expert training or experience. Underestimating the severity of the disease is potentially a health risk, given that the severity of colonic inflammation is an important determinant in the risk of colorectal neoplasia in long-standing extensive UC [19]. As such, transmural impedance might act as a useful, objective, spatially selective biomarker to categorize disease activity of IBD patients during an endoscopic examination.

Mucosal impedance (commonly reported as admittance, the inverse of impedance), has been used for real-time evaluation of mucosal integrity in a number of gastrointestinal diseases. The prognosis of UC patients in remission was accurately predicted from measuring changes in admittance [11]. Furthermore, measuring trans-mucosal admittance accurately distinguished gastroesophageal reflux (GERD) from non-GERD diseases such as heart burn [20], and aided in the diagnosis of duodenal dyspepsia [21].

The current study was limited in that it only related impedance measurements to histological score and inflammatory cells counts within regional areas of inflamed tissue. It would have been informative to also perform an endoscopic assessment and correlate this to the impedance and histology. Furthermore, as a future study, using dextran sulfate sodium to induce chronic or relapsing colitis [22] would be important in determining if the transmural impedance is a useful measurement of graded temporal changes to mucosal integrity. In conclusion, the ability to accurately diagnose UC is critical to managing the disease. In this study, we conclude that real-time measurement of transmural impedance is a spatially specific indicator of graded levels of mucosal integrity.

Data accessibility. Data is available in an external publically shared folder in the Dryad Digital Repository at: https://doi.org/10.5061/dryad.79cnp5hrb [23].

Authors' contributions. S.C.P. conducted the experiments, performed the histopathological scoring of tissue, analysed the data and wrote the manuscript; J.A. conducted the experiments and assisted with data analysis; R.T. was involved in generating the concept and acquiring data; R.K.S was involved with interpretation of data and manuscript drafting; J.B.Fu. contributed to the design of the research and manuscript drafting; J.B.Fa. gave intellectual input on experimental design, data meaning and interpretation and manuscript writing revisions. All authors gave final approval for publication.

Competing interests. We declare we have no competing interests.

Funding. This work was sponsored by the Defense Advanced Research Projects Agency (DARPA) BTO under the auspices of Dr Doug Weber and Dr Eric Van Gieson through the Space and Naval Warfare Systems Center (contract no. N66001-15-2-4060). The Bionics Institute and the Florey Institute of Neuroscience and Mental Health acknowledge the support they receive from the Victorian Government through its Operational Infrastructural Support Program.

Acknowledgements. The authors are grateful to Remy Constable and Tomoko Hyakumura for histological assistance.

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
