## [Reviewer comments · Royal Society Open Science]

Review History

RSOS-191819.R0 (Original submission)

Review form: Reviewer 1

Is the manuscript scientifically sound in its present form?

Yes

Are the interpretations and conclusions justified by the results?

Yes

Is the language acceptable?

Yes

Do you have any ethical concerns with this paper?

No

Have you any concerns about statistical analyses in this paper?

No

Recommendation?

Accept as is

Comments to the Author(s)

Great piece of work by the authors. Due to the previous publications by the authors themselves in the small intestine and a clinical study by another group, it is not surprising that the intestinal transmural impedance correlates with histological assessment of mucosal damage in the colon, but this is still a necessary contribution to literature. Indeed there are limitations to the study as the authors pointed out, but this can be addressed in future work.

Review form: Reviewer 2**Is the manuscript scientifically sound in its present form?**

Yes

Are the interpretations and conclusions justified by the results?

No

Is the language acceptable?

Yes

Do you have any ethical concerns with this paper?

No

Have you any concerns about statistical analyses in this paper?

Yes

Recommendation?

Major revision is needed (please make suggestions in comments)

Comments to the Author(s)

The manuscripts describes transmural impedance measurements as an indicator for intestinal inflammation. This article is a consequent follow-up study and in vivo translation as compared to previous work. However, the current manuscript has quite significant overlap with a past publication from the same group.

In its current form it raises some concerns:

Major comments:

- It is not clear how the authors created the cut-off values for different grades of inflammation. How are different grades of inflammation as measured by impedance values statistically justified? How are ranges defined? (see abstract p.2 l.39-41) The "n" is quite small to justify this finding.
- Typically one would expect a sham control experiment. How robust is your approach in a control animal treatment with ethanol only? And along that line, is the cell infiltration the cause of impedance change or is it the direct damage to the epithelial layer (caused by ethanol)? Taken it further, would this approach also measure flares of inflammation in chronically inflamed/fibrotic tissues (which would be clinically more relevant)? This would not also be feasible in a DSS model but also with repeated doses of TNBS.

Minor comments:

Introduction:

- Please shorten introduction.

- There is a critical statement in the introduction (p.3 l.53-56): Indication for surgery in UC include: uncontrolled hemorrhage, perforation, refractory acute severe or medically refractory disease, and colorectal carcinoma or dysplastic lesions. Please clarify.

Material and Methods:

- Please clearly state how many rats served as experimental and how many as control (4vs4).
- Please give weight of experimental animals
- p.7 l.152 "A single observer (S. C. Payne)..." please confirm that this statement is in accordance with your author contributions. Was this procedure done (blinded) by some else or the first author?

Discussion:

- Requires shortening

Statistics:

- Please check your statistical testing. I would assume 5.5 and 8cm measurements (within the same subject) are not independent measurements.

Table 1 could be moved to appendix

References:

Payne, S. C., R. K. Shepherd, A. Sedo, J. B. Fallon and J. B. Furness (2018). "An objective in vivo diagnostic method for inflammatory bowel disease." *R Soc Open Sci* 5(3): 180107.

Ungaro, R., S. Mehandru, P. B. Allen, L. Peyrin-Biroulet and J. F. Colombel (2017). "Ulcerative colitis." *Lancet* 389(10080): 1756-1770.

Fichtner-Feigl, S., I. J. Fuss, C. A. Young, T. Watanabe, E. K. Geissler, H. J. Schlitt, A. Kitani and W. Strober (2007). "Induction of IL-13 triggers TGF-beta1-dependent tissue fibrosis in chronic 2,4,6-trinitrobenzene sulfonic acid colitis." *J Immunol* 178(9): 5859-5870.

Decision letter (RSOS-191819.R0)

20-Jan-2020

Dear Dr Payne,

The editors assigned to your paper ("Transmural impedance detects graded changes of inflammation in experimental colitis") have now received comments from reviewers. We would like you to revise your paper in accordance with the referee and Associate Editor suggestions which can be found below (not including confidential reports to the Editor). Please note this decision does not guarantee eventual acceptance.

Please submit a copy of your revised paper before 12-Feb-2020. Please note that the revision deadline will expire at 00.00am on this date. If we do not hear from you within this time then it will be assumed that the paper has been withdrawn. In exceptional circumstances, extensions may be possible if agreed with the Editorial Office in advance. We do not allow multiple rounds of revision so we urge you to make every effort to fully address all of the comments at this stage. If deemed necessary by the Editors, your manuscript will be sent back to one or more of the

original reviewers for assessment. If the original reviewers are not available, we may invite new reviewers.

- Data accessibility

If you wish to submit your supporting data or code to Dryad (<http://datadryad.org/>), or modify your current submission to dryad, please use the following link:
<http://datadryad.org/submit?journalID=RSOS&manu=RSOS-191819>

- Competing interests

- Authors' contributions

AB carried out the molecular lab work, participated in data analysis, carried out sequence alignments, participated in the design of the study and drafted the manuscript; CD carried out

the statistical analyses; EF collected field data; GH conceived of the study, designed the study, coordinated the study and helped draft the manuscript. All authors gave final approval for publication.

- Acknowledgements

- Funding statement

Best regards,

on behalf of Dr Andy Greenfield (Associate Editor) and Kevin Padian (Subject Editor)
openscience@royalsociety.org

Associate Editor's comments (Dr Andy Greenfield):

Thanks you for submitting. We have now received the opinions of two expert reviewers and as you will see, one of these makes significant criticisms of your manuscript. We would like to invite you to respond to these, paying particular attention to comments concerning the how cut-off values are defined and statistical justification, and in particular the absence of a sham control. If you are able to respond to these, and the other, comments to our satisfaction by revising your manuscript, we would be willing to consider publishing it in RSOS

Reviewers' Comments to Author:

Reviewer: 1

Comments to the Author(s)

Great piece of work by the authors. Due to the previous publications by the authors themselves in the small intestine and a clinical study by another group, it is not surprising that the intestinal transmural impedance correlates with histological assessment of mucosal damage in the colon, but this is still a necessary contribution to literature. Indeed there are limitations to the study as the authors pointed out, but this can be addressed in future work.

Reviewer: 2

Comments to the Author(s)

The manuscripts describes transmural impedance measurements as an indicator for intestinal inflammation. This article is a consequent follow-up study and in vivo translation as compared to previous work. However, the current manuscript has quite significant overlap with a past publication from the same group.

In its current form it raises some concerns:

Major comments:

- It is not clear how the authors created the cut-off values for different grades of inflammation. How are different grades of inflammation as measured by impedance values statistically justified? How are ranges defined? (see abstract p.2 1.39-41) The “n” is quite small to justify this finding.
- Typically one would expect a sham control experiment. How robust is your approach in a control animal treatment with ethanol only? And along that line, is the cell infiltration the cause of impedance change or is it the direct damage to the epithelial layer (caused by ethanol)? Taken it further, would this approach also measure flares of inflammation in chronically inflamed/fibrotic tissues (which would be clinically more relevant)? This would not also be feasible in a DSS model but also with repeated doses of TNBS.

Minor comments:

Introduction:

- Please shorten introduction.
- There is a critical statement in the introduction (p.3 1.53-56): Indication for surgery in UC include: uncontrolled hemorrhage, perforation, refractory acute severe or medically refractory disease, and colorectal carcinoma or dysplastic lesions. Please clarify.

Material and Methods:

- Please clearly state how many rats served as experimental and how many as control (4vs4).
- Please give weight of experimental animals
- p.7 1.152 “A single observer (S. C. Payne)...” please confirm that this statement is in accordance with your author contributions. Was this procedure done (blinded) by some else or the first author?

Discussion:

- Requires shortening

Statistics:

- Please check your statistical testing. I would assume 5.5 and 8cm measurements (within the same subject) are not independent measurements.

Table 1 could be moved to appendix

References:

Payne, S. C., R. K. Shepherd, A. Sedo, J. B. Fallon and J. B. Furness (2018). "An objective in vivo diagnostic method for inflammatory bowel disease." *R Soc Open Sci* 5(3): 180107.

Ungaro, R., S. Mehandru, P. B. Allen, L. Peyrin-Biroulet and J. F. Colombel (2017). "Ulcerative colitis." *Lancet* 389(10080): 1756-1770.

Fichtner-Feigl, S., I. J. Fuss, C. A. Young, T. Watanabe, E. K. Geissler, H. J. Schlitt, A. Kitani and W. Strober (2007). "Induction of IL-13 triggers TGF-beta1-dependent tissue fibrosis in chronic 2,4,6-trinitrobenzene sulfonic acid colitis." *J Immunol* 178(9): 5859-5870.

Author's Response to Decision Letter for (RSOS-191819.R0)

See Appendix A.

Decision letter (RSOS-191819.R1)

27-Jan-2020

Dear Dr Payne,

It is a pleasure to accept your manuscript entitled "Transmural impedance detects graded changes of inflammation in experimental colitis" in its current form for publication in Royal Society Open Science. The comments of the reviewer(s) who reviewed your manuscript are included at the foot of this letter.

on behalf of Dr Andy Greenfield (Associate Editor) and Kevin Padian (Subject Editor)
openscience@royalsociety.org

Appendix A

Dr. Andy Greenfield,
Associate Editor *Royal Society Open Science*

Dear Dr. Greenfield,

I am writing to re-submit the manuscript RSOS-191819 entitled “*Transmural impedance detects graded changes of inflammation in experimental colitis*” to the Royal Society Open Science. This manuscript is not under consideration for publication in any other journal, and all authors have approved the revisions to the manuscript and its resubmission to the journal.

We thank the Reviewers for their comments and have revised the manuscript accordingly. Our responses to the comments raised by the reviewers are described below, and we have tracked all changes in the manuscript.

Regards,

Sophie Payne
Research Fellow
Bionics Institute

384 Albert Street
East Melbourne 3002
Mob: +61 404 830372
Web: www.bionicsinstitute.org/our-staff/Pages/SophiePayne.aspx

Reviewer 1

“Great piece of work by the authors. Due to the previous publications by the authors themselves in the small intestine and a clinical study by another group, it is not surprising that the intestinal transmural impedance correlates with histological assessment of mucosal damage in the colon, but this is still a necessary contribution to literature. Indeed there are limitations to the study as the authors pointed out, but this can be addressed in future work.”

We thank Reviewer one for this comment.

Reviewer 2

Major comments

1. *The manuscript describes transmural impedance measurements as an indicator for intestinal inflammation. This article is a consequent follow-up study and in vivo translation as compared to previous work. However, the current manuscript has quite significant overlap with a past publication from the same group.*

We believe that this work builds on our previous work (Payne et al., 2018 R Soc Open Sci), rather than overlaps it. The Payne 2018 study shows that normalised electrical impedance could serve as an objective marker of mucosal integrity following TNBS-induced inflammation in the small intestine of rat, but it used a highly invasive approach

In the current study, we built on our previous work by demonstrating that absolute electrical impedance serves as an objective marker of mucosal integrity following TNBS-induced inflammation of the colon. Unlike the previous study, the electrode was minimally invasive and introduced via the rectum, an approach that has translational potential for clinical use, during for example a colonoscopy.

The current study also assessed whether the changes in transmucosal impedance were spatially restricted to the site of inflammation and therefore able to determine both the site and the severity of intestinal inflammation. We predict that having the ability to distinguish between different levels of inflammation within the gastrointestinal tract will support the use of this diagnostic tool during an endoscopic examination. We have included this discussion point in the discussion (lines 303-307).

2. *“It is not clear how the authors created the cut-off values for different grades of inflammation. How are different grades of inflammation as measured by impedance values statistically justified?”*

The different grades of inflammation are determined using a histological scoring system adapted from Erben et al., 2014. Int J Clin Exp Pathol. 7: 8: 4557-76. In this reference, a score of ‘0’ for a particular histopathology relates to minimal inflammatory damage (i.e. ‘normal’), ‘1’ for mild damage, ‘2’ for moderate damage and ‘3’ for marked/severe damage. The different grades of minimally inflamed, mildly inflamed and moderately inflamed gradings are subjective, but they are based on established standards for scoring histopathological inflammatory-induced damage. Note the degrees of inflammation are based on the histological assessment, and not the impedance values, and therefore there is no ‘statistical justification’ of the grades as measured by the impedance. However, in the results section we have included the range and standard deviation of impedances generated from tissue with minimal, mild and moderate inflammation (lines 181-183; 193-196).

In-line with the above comment, we have revised the use of the word ‘normal’ (line 40, page 2) to describe tissue that scored between 0-2 out of 15 and have changed in the manuscript to be ‘minimally inflamed’ (lines 32, 40, 97, 207, 227, 232, 298, 301).

- a. *How are ranges defined? (see abstract p.2 l.39-41) The “n” is quite small to justify this finding.”*

The range of impedance values refers to the minimum and maximum impedance measurement generated from tissue that had minimal, mild or moderate levels of inflammatory damage. We have revised the text to make this clearer (line 40).

Using an 'n' of 4 normal rats and 4 experimental rats resulted in statistically significant differences in impedances between grades of inflamed tissue, which supports our conclusions.

3. *“Typically one would expect a sham control experiment. How robust is your approach in a control animal treatment with ethanol only?”*

The inflammatory inducing TNBS injection is made up in 50% ethanol, which is used to disrupt the epithelial cell barrier in order to facilitate the entry of TNBS into the sub-epithelial layers. As such, ethanol is an integral part of the inflammatory challenge, and we believe that using ethanol as a sham (control) injection is not appropriate. Instead, we injected sterile saline as a vehicle control into normal animals prior to taking impedance measurements. This detail has been clarified in the manuscript (lines 110-111). Furthermore, as our aim was to correlate impedance and histological change, investigating the effects of an ethanol only insult was not required.

a. *And along that line, is the cell infiltration the cause of impedance change or is it the direct damage to the epithelial layer (caused by ethanol)?*

Impedance change is likely to be affected by both the disruption to the epithelial barrier from ethanol as well as from the activity of immune cell infiltration, as immune cells (such as neutrophils and eosinophils) secrete pro-inflammatory cytokines that activate a cascade of intracellular mechanisms that lead to the disruption of tight junctions, which regulate paracellular permeability of the epithelial cell barrier (Suzuki 2013. Cell Mol Life Sci. DOI: 10.1007/s00018-012-1070-x).

b. *Taken it further, would this approach also measure flares of inflammation in chronically inflamed/fibrotic tissues (which would be clinically more relevant)? This would not also be feasible in a DSS model but also with repeated doses of TNBS.”*

We assume that the principles of detecting inflammation in this acute study will apply to chronically inflamed tissue. However, it is a focus of subsequent studies to further examine the validity of this objective measurement in a chronic model of colitis (such as DSS). We agree with the Reviewer that repeated doses of TNBS to induce remitting colitis could also be a useful model of chronic inflammation.

Minor comments

1. *Introduction:*

a. *- Please shorten introduction.*

We have shortened the introduction by about 20%.

b. *There is a critical statement in the introduction (p.3 l.53-56): Indication for surgery in UC include: uncontrolled hemorrhage, perforation, refractory*

acute severe or medically refractory disease, and colorectal carcinoma or dysplastic lesions. Please clarify.

We have included some of these symptoms that often lead to colectomy (lines 54). The text now reads: *“The debilitating symptoms of the disease include weight loss, diarrhoea, hemorrhage, perforation and dysplastic lesions that may lead to a colectomy in some patients”.*

2. *Material and Methods:*

- a. *Please clearly state how many rats served as experimental and how many as control (4vs4).*

There were 4 rats used as control and 4 rats that received experimental colitis. We have clarified this in the methods (lines 109, 113).

- b. *Please give weight of experimental animals*

We have given a weight range of animals (350-370 g) used in this study (lines 104).

- c. *p.7 l.152 “A single observer (S. C. Payne)...” please confirm that this statement is in accordance with your author contributions. Was this procedure done (blinded) by some else or the first author?*

We have added histopathological scoring of tissue to S.C.Payne’s author’s contributions (lines 347-348). This procedure was conducted blinded by the first author (S.C.Payne).

3. *Discussion:*

- a. *Requires shortening*

We have shortened the discussion by about 10% (lines 314-316).

4. *Statistics:*

- a. *Please check your statistical testing. I would assume 5.5 and 8cm measurements (within the same subject) are not independent measurements.*

We agree with the author that the analysis between 5.5 cm and 8 cm tissue are dependent. However, the analysis performed for statistic displayed in Figures 2 and 3 were between tissue taken (from 5.5 cm and 8 cm regions) from normal vs. inflamed rats, which are independent samples. As such, a one-way ANOVA was conducted for this analysis. This has been clarified in the methods (line 172). Furthermore, the statistical test used to test difference between 5.5 cm and 8 cm was a paired t-test (line 184).

5. *Table 1 could be moved to appendix*

We would prefer for Table 1 to be included in the main text as this is an important method.

6. *References*

The DOI has been added to these references.